# The Mechanistic Emergence of Symbol Grounding in Language Models

**Shuyu Wu** [* 1]  **Ziqiao Ma** [* 1]  **Xiaoxi Luo** [* 2 3]  **Yidong Huang** [1 4]  **Josue Torres-Fonseca** [1]
**Freda Shi** [† 2 3]  **Joyce Chai** [† 1]

## Abstract

Symbol grounding (Harnad, 1990) describes how symbols such as words acquire their meanings by connecting to real-world sensorimotor experiences. Recent work has shown preliminary evidence that grounding may emerge in (vision-)language models trained at scale without using explicit grounding objectives. Yet, the specific loci of this emergence and the mechanisms that drive it remain largely unexplored. To address this problem, we introduce a controlled evaluation framework that systematically traces how symbol grounding arises within the internal computations through mechanistic and causal analysis. Our findings show that grounding concentrates in middle-layer computations and is implemented through the aggregate mechanism, where attention heads aggregate the environmental ground to support the prediction of linguistic forms. This phenomenon replicates in multimodal dialogue and across architectures (Transformers and state-space models), but not in unidirectional LSTMs. Our results provide behavioral and mechanistic evidence that symbol grounding can emerge in language models, with practical implications for predicting and potentially controlling the reliability of generation.

## 1. Introduction

Symbol grounding (Harnad, 1990) refers to the problem of how abstract and discrete symbols, such as words, acquire meaning by connecting to perceptual or sensorimotor experiences. In the context of multimodal machine learning, grounding has served as a pre-training objective for vision-language models (VLMs), by connecting linguistic units to the world that gives language meanings (Li et al., 2022; Ma et al., 2023). Through supervised fine-tuning with explicit grounding supervision such as entity-phrase mappings, modern VLMs have achieved fine-grained understanding at both region (You et al., 2024; Peng et al., 2024; Wang et al., 2024) and pixel (Zhang et al., 2024b; Rasheed et al., 2024; Zhang et al., 2024a) levels.

Meanwhile, with the rising of autoregressive language models (LMs; OpenAI, 2024; Anthropic, 2024; Comanici et al., 2025, *inter alia*) and their VLM extensions, there is growing interest in identifying and interpreting their emergent capabilities. Recent work has shown preliminary correlational evidence that grounding may emerge in LLMs (Sabet et al., 2020; Shi et al., 2021; Wu et al., 2025) and VLMs (Cao et al., 2024; Bousselham et al., 2024; Schnaus et al., 2025) trained at scale, even when solely optimized with the simple next-token prediction objective. However, the underlying mechanisms that lead to such an emergence are not well understood. To address this limitation, our work seeks to understand the emergence of symbol grounding in multimodal language models, causally and mechanistically tracing how symbol grounding arises within the internal computations.

Given the intractability of directly interpreting large, complex VLMs, we begin by constructing a minimal testbed to investigate grounding mechanisms in a synthetic multimodal setting. Our testbed is derived from the CHILDES corpora (MacWhinney, 2000), which provide cognitively plausible contexts and utterances for human language acquisition research. In our framework, each word is represented in two distinct forms: one token that appears in non-verbal scene descriptions (e.g., *box* in the environment description) and another that appears in spoken utterances (e.g., *box* in dialogue). We refer to these as environmental tokens ($\langle$ENV$\rangle$) and linguistic tokens ($\langle$LAN$\rangle$), respectively. A deliberately simple word-level tokenizer assigns separate vocabulary indices to each form, ensuring that they are treated as entirely different tokens by the language model. This structural separation prevents grounding from being reduced to the trivial case of shared token identity, and offers a controlled, noise-free proxy for studying the grounding mechanisms. We use this testbed to derive hypotheses, which are subsequently validated with realistic VLMs.

---

[*]Equal contribution  [1]University of Michigan [2]University of Waterloo [3]Vector Institute [4]University of North Carolina at Chapel Hill. Correspondence to: Shuyu Wu <shuyuwu@umich.edu>, Freda Shi <fhs@uwaterloo.ca>, Joyce Chai <chaijy@umich.edu>.

*Proceedings of the 43rd International Conference on Machine Learning*, Seoul, South Korea. PMLR 306, 2026. Copyright 2026 by the author(s).

With the testbed, we quantify grounding using surprisal: specifically, we compare how easily the model predicts a linguistic token ($\langle$LAN$\rangle$) when its matching environmental token ($\langle$ENV$\rangle$) is present versus when unrelated cues are given instead. A lower surprisal in the former condition indicates that the model has learned to align environmental grounds with linguistic forms. We find that LMs do learn to ground: the presence of environmental tokens consistently reduces surprisal for their linguistic counterparts, in a way that simple co-occurrence statistics cannot fully explain. To study the underlying mechanisms, we apply saliency analysis (Wang et al., 2023) and the tuned lens (Belrose et al., 2023), which converge on the result that grounding relations are concentrated in the middle layers of the network. Further analysis of attention heads reveals patterns consistent with the aggregate mechanism (Bick et al., 2025), where attention heads support the prediction of linguistic forms by retrieving their environmental grounds in the context. Figure 1a and 1c illustrates this pattern with an example.

Finally, we demonstrate that these findings generalize beyond the minimal testbed with CHILDES data and Transformer models. We observe that similar trends appear in a multimodal setting with the Visual Dialog dataset (Das et al., 2017), and in state-space models (SSMs) such as Mamba-2 (Dao & Gu, 2024). In contrast, we do not observe grounding in unidirectional LSTMs, consistently with their sequential state compression and lack of content-addressable retrieval. Taken together, our results show that symbol grounding can mechanistically emerge in autoregressive multimodal LMs, as well as delineating the architectural conditions under which it can arise.

## 2. Related Work

**Language grounding.** Referential grounding has long been framed as the lexicon acquisition problem: how words map to referents in the world (Harnad, 1990; Gleitman & Landau, 1994; Clark, 1995). Early work focused on word-to-symbol mappings, designing learning mechanisms that simulate children's lexical acquisition and explain psycholinguistic phenomena (Siskind, 1996; Regier, 2005; Goodman et al., 2007; Fazly et al., 2010). Subsequent studies incorporated visual grounding, first by aligning words with object categories (Roy & Pentland, 2002; Yu, 2005; Xu & Tenenbaum, 2007; Yu & Ballard, 2007; Yu & Siskind, 2013), and later by mapping words to richer visual features (Qu & Chai, 2010; Mao et al., 2019; 2021; Pratt et al., 2020). More recently, large-scale VLMs trained with paired text–image supervision have advanced grounding to finer levels of granularity, achieving region-level (Li et al., 2022; Ma et al., 2023; Chen et al., 2023; You et al., 2024; Wang et al., 2024) and pixel-level (Xia et al., 2024; Rasheed et al., 2024; Zhang et al., 2024b) grounding, with strong performance on referring expression comprehension (Chen et al., 2024a).

Recent work suggests that grounding emerges as a property of VLMs trained without explicit supervision, with evidence drawn from attention-based spatial localization (Cao et al., 2024; Bousselham et al., 2024) and cross-modal geometric correspondences (Schnaus et al., 2025). However, all prior work focused exclusively on static final-stage models, overlooking the training trajectory, a crucial aspect for understanding when and how grounding emerges. In addition, existing work has framed grounding through correlations between visual and textual signals, diverging from the definition by Harnad (1990), which emphasizes causal links from symbols to meanings. To address these issues, we systematically examine learning dynamics throughout model training, applying causal interventions to probe model internals and introducing control groups to enable rigorous comparison.

**Emergent capabilities and learning dynamics of LMs.** A central debate concerns whether larger language models exhibit genuinely new behaviors: Wei et al. (2022) highlight abrupt improvements in tasks, whereas later studies argue such effects are artifacts of thresholds or in-context learning dynamics (Schaeffer et al., 2023; Lu et al., 2023). Beyond end performance, developmental analyses show that models acquire linguistic abilities in systematic though heterogeneous orders with variability across runs and checkpoints (Sellam et al., 2021; Blevins et al., 2022; Biderman et al., 2023; Xia et al., 2023; van der Wal et al., 2025). Psychology-inspired perspectives further emphasize controlled experimentation to assess these behaviors (Hagendorff, 2023), and comparative studies reveal both parallels and divergences between machine and human language learning (Chang & Bergen, 2022; Evanson et al., 2023; Chang et al., 2024; Ma et al., 2025). At a finer granularity, hidden-loss analyses identify phase-like transitions (Kangaslahti et al., 2025), while distributional studies attribute emergence to stochastic differences across training seeds (Zhao et al., 2025). Together, emergent abilities are not sharp discontinuities but probabilistic outcomes of developmental learning dynamics. Following this line, we present a probability- and model internals–based analysis of how symbol grounding emerges during language model training.

**Mechanistic interpretability of LMs.** Mechanistic interpretability has largely focused on attention heads in Transformers (Elhage et al., 2021; Olsson et al., 2022; Meng et al., 2023; Bietti et al., 2023; Lieberum et al., 2023; Wu et al., 2024). A central line of work established that *induction heads* emerge to support in-context learning (ICL; Elhage et al., 2021; Olsson et al., 2022), with follow-up studies tracing their training dynamics (Bietti et al., 2023) and mapping factual recall circuits (Meng et al., 2023). At larger scales, Lieberum et al. (2023) identified specialized *content-gatherer* and *correct-letter* heads, and Wu et al. (2024) showed that a sparse set of *retrieval heads* is critical for reasoning and long-context performance. Relatedly,

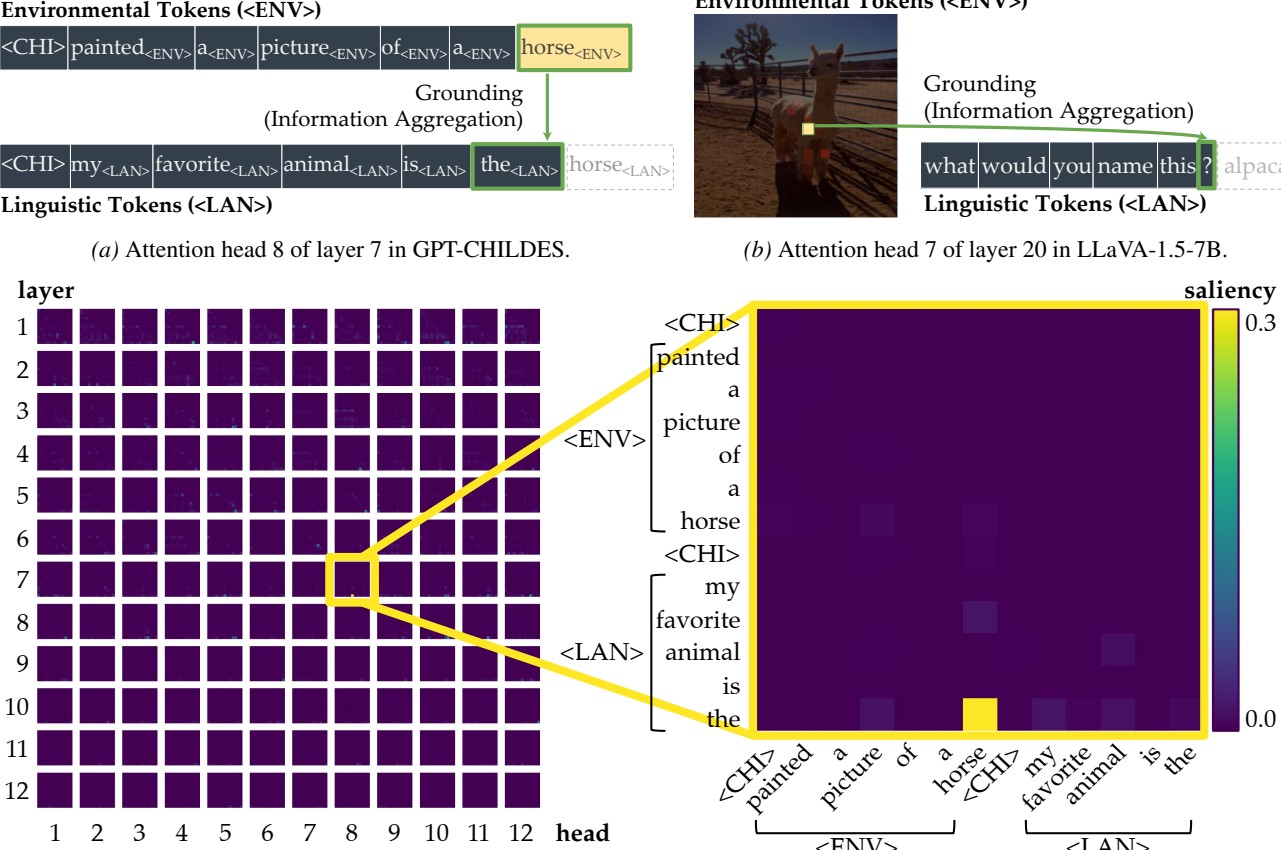

**Environmental Tokens (<ENV>)**

<CHI> painted<ENV> a<ENV> picture<ENV> of<ENV> a<ENV> horse<ENV>

Grounding
(Information Aggregation)

<CHI> my<LAN> favorite<LAN> animal<LAN> is<LAN> the<LAN> horse<LAN>

**Linguistic Tokens (<LAN>)**

*(a)* Attention head 8 of layer 7 in GPT-CHILDES.

**Environmental Tokens (<ENV>)**

Grounding
(Information Aggregation)

what would you name this ? alpaca

**Linguistic Tokens (<LAN>)**

*(b)* Attention head 7 of layer 20 in LLaVA-1.5-7B.

*(c)* Left: saliency over tokens of each head in each layer for the prompt $\langle CHI\rangle$ $painted_{\langle ENV\rangle}$ $a_{\langle ENV\rangle}$ $picture_{\langle ENV\rangle}$ $of_{\langle ENV\rangle}$ $a_{\langle ENV\rangle}$ $horse_{\langle ENV\rangle}$ $\langle CHI\rangle$ $my_{\langle LAN\rangle}$ $favorite_{\langle LAN\rangle}$ $animal_{\langle LAN\rangle}$ $is_{\langle LAN\rangle}$ $the_{\langle LAN\rangle}$. Right: among all, only one of them (head 8 of layer 7) is identified as an aggregate head, where information flows from $horse_{\langle ENV\rangle}$ to the current position, encouraging the model to predict $horse_{\langle LAN\rangle}$ as the next token.

*Figure 1.* Illustration of the symbol grounding mechanism through information aggregation. Lighter colors denote more salient attention, quantified by saliency scores, i.e., gradient × attention contributions to the loss (Wang et al., 2023). When predicting the next token, aggregate heads (Bick et al., 2025) emerge to exclusively link environmental tokens (visual or situational context; $\langle ENV\rangle$) to linguistic tokens (words in text; $\langle LAN\rangle$). These heads provide a mechanistic pathway for symbol grounding by mapping external environmental evidence into its linguistic form.

Wang et al. (2023) demonstrated that label words in demonstrations act as *anchors*: early layers gather semantic information into these tokens, which later guide prediction. Based on these insights, Bick et al. (2025) proposed that retrieval is implemented through a coordinated *gather-and-aggregate (G&A)* mechanism: some heads collect content from relevant tokens, while others aggregate it for prediction. Other studies extended this line of work by analyzing failure modes and training dynamics (Wiegreffe et al., 2025) and contrasting retrieval mechanisms in Transformers and SSMs (Arora et al., 2025). Whereas prior analyses typically investigate ICL with repeated syntactic or symbolic formats, our setup requires referential alignment between linguistic forms and their environmental contexts, providing a complementary testbed for naturalistic language grounding.

## 3. Method

### 3.1. Dataset and Tokenization

To capture the emergent grounding in multimodal interactions, we design a minimal testbed with a custom word-level tokenizer, in which every lexical item is represented in two corresponding forms: one token that appears in non-verbal descriptions (e.g., a *book* in the scene description) and another that appears in utterances (e.g., *book* in speech). We refer to these by environmental ($\langle ENV\rangle$) and linguistic tokens ($\langle LAN\rangle$), respectively. For instance, $book_{\langle ENV\rangle}$ and $book_{\langle LAN\rangle}$ receive different integer indices from the tokenizer; that is, tokenization provides no explicit signal that these tokens are related, so any correspondence between them must be learned during training rather than inherited from surface forms. Although our minimal testbed uses child-directed

speech data, the framework is readily extensible to visual dialogue datasets, enabling the investigation of VLMs in realistic settings.

**Child-directed speech.** The Child Language Data Exchange System (CHILDES; MacWhinney, 2000) provides transcripts enriched with environmental annotations.[1] We use the spoken utterances as the linguistic tokens ($\langle$LAN$\rangle$) and the environmental descriptions as the environment tokens ($\langle$ENV$\rangle$). The environmental context is drawn from three annotation types:

- **Local events**: simple events, pauses, long events, or remarks interleaved with the transcripts.

- **Action tiers**: actions performed by the speaker or listener (e.g., `%act: runs to toy box`). These also include cases where an action replaces speech (e.g., `0 [% kicks the ball]`).

- **Situational tiers**: situational information tied to utterances or to larger contexts (e.g., `%sit: dog is barking`).

**Image-grounded dialogue.** To move beyond the minimal textual proxies, we create an image-grounded dialogue setup using the Visual Dialog dataset (Das et al., 2017), which pairs MSCOCO images (Lin et al., 2014) with sequential multi-turn question-answering dialogues that exchange information about each image. Here, a frozen vision transformer (ViT; Dosovitskiy et al., 2020) converts an RGB image into patch embeddings, with each embedding treated as an $\langle$ENV$\rangle$ token, analogously to the visual tokens in modern VLMs.[2] We use DINOv2 (Oquab et al., 2024) as our ViT tokenizer—because it is trained exclusively on vision data without auxiliary text supervision (unlike CLIP; Radford et al., 2021), it ensures that environmental tokens encode strictly visual information. The dialogues correspond to linguistic tokens ($\langle$LAN$\rangle$), forming realistic multimodal interactions where conversational utterances are grounded directly in visual input.

We also introduce an intermediate setup: caption-grounded visual dialogue, which uses MSCOCO image captions as the grounding context. This configuration bridges the gap between the synthetic child-directed speech environment and the realistic image-grounded dialogue settings. Further details can be found in Appendix C.1.

### 3.2. Evaluation Protocol

We assess symbol grounding with a contrastive test that asks whether a model assigns a higher probability to the correct linguistic token when the matching environmental token is

in context, following the idea of priming in psychology. This evaluation applies uniformly across datasets (Table 1): in CHILDES, environmental priming comes from descriptive contexts; in image-grounded dialogue, from ViT-derived visual tokens. We compare the following conditions:

- **Match (experimental condition)**: The context contains the corresponding $\langle$ENV$\rangle$ token for the target word, and the model is expected to predict its $\langle$LAN$\rangle$ counterpart.

- **Mismatch (control condition)**: The context is replaced with a different $\langle$ENV$\rangle$ token. The model remains tasked with predicting the same $\langle$LAN$\rangle$ token; however, in the absence of corresponding environmental cues, its performance is expected to be no better than chance.

For example (first row in Table 1), when evaluating the word $book_{\langle \text{LAN} \rangle}$, the input context is

$$\langle CHI \rangle \ asked_{\langle \text{ENV} \rangle} \ for_{\langle \text{ENV} \rangle} \ a_{\langle \text{ENV} \rangle} \ new_{\langle \text{ENV} \rangle} \ book_{\langle \text{ENV} \rangle}$$
$$\langle CHI \rangle \ I_{\langle \text{LAN} \rangle} \ love_{\langle \text{LAN} \rangle} \ this_{\langle \text{LAN} \rangle} \ \underline{\quad\quad}, \tag{1}$$

where the model is expected to predict $book_{\langle \text{LAN} \rangle}$ for the blank, and the role token $\langle CHI \rangle$ indicates the involved speaker or actor's role being a child. In the control (mismatch) condition, the environmental token $box_{\langle \text{ENV} \rangle}$ is replaced by another valid noun such as $toy_{\langle \text{ENV} \rangle}$.

**Context templates.** For a target word $v$ with linguistic token $v_{\langle \text{LAN} \rangle}$ and environmental token $v_{\langle \text{ENV} \rangle}$, we denote $\overline{C}_v$ as a set of context templates of $v$. For example, when $v = book$, a $\overline{c} \in \overline{C}_v$ can be

$$\langle CHI \rangle \ asked_{\langle \text{ENV} \rangle} \ for_{\langle \text{ENV} \rangle} \ a_{\langle \text{ENV} \rangle} \ new_{\langle \text{ENV} \rangle} \ [\text{FILLER}]$$
$$\langle CHI \rangle \ I_{\langle \text{LAN} \rangle} \ love_{\langle \text{LAN} \rangle} \underline{\quad\quad}, \tag{2}$$

where [FILLER] is to be replaced with an environmental token, and the blank indicates the expected prediction as in Eq. (1). In the match condition, the context $\overline{c}(v)$ is constructed by replacing [FILLER] with $v_{\langle \text{ENV} \rangle}$ in $\overline{c}$. In the mismatch condition, the context $\overline{c}(u)$ uses $u_{\langle \text{ENV} \rangle}$ ($u \neq v$) as the filler, while the prediction target remains $v_{\langle \text{LAN} \rangle}$.

For the choices of $v$ and $u$, we construct the vocabulary $V$ with 100 nouns from the MacArthur–Bates Communicative Development Inventories (Fenson et al., 2006) that occur frequently in our corpus. Each word serves once as the target, with the remaining $M = 99$ used to construct mismatched conditions. For each word, we create $N = 10$ context templates, which contain both $\langle$ENV$\rangle$ and $\langle$LAN$\rangle$ tokens. Details of the vocabulary and context template construction can be found in the Appendix A.

**Grounding information gain.** Following prior work, we evaluate how well an LM learns a word using the mean surprisal over instances. The surprisal of a word $w$ given a context $c$ is defined as $s_{\boldsymbol{\theta}}(w \mid c) = -\log P_{\boldsymbol{\theta}}(w \mid c)$, where $P_{\boldsymbol{\theta}}(w \mid c)$ denotes the probability, under an LM

---

[1]See the manual for data usage: https://talkbank.org/0info/manuals/CHAT.pdf

[2] Note that these "visual tokens" are continuously valued and do not correspond to discrete symbol-like tokens. They, as a whole, can be considered tensor representations of an image.

*Table 1.* Training and test examples across datasets with target word *book*. The training examples combine environmental tokens ( ⟨ENV⟩; shaded ) with linguistic tokens (⟨LAN⟩). Test examples are constructed with either matched (*book*) or mismatched (*toy*) environmental contexts, paired with corresponding linguistic prompts. Note that in child-directed speech, $book_{⟨ENV⟩}$ and $book_{⟨LAN⟩}$ are two distinct tokens received by LMs.

| Dataset | Training Example | | Test Example | | |
|---|---|---|---|---|---|
| | ⟨ENV⟩ | ⟨LAN⟩ | ⟨ENV⟩ Match | ⟨ENV⟩ Mismatch | ⟨LAN⟩ |
| **Child-Directed Speech** | ⟨CHI⟩ takes book from mother | ⟨CHI⟩ what's that ⟨MOT⟩ a book in it ... | ⟨CHI⟩ asked for a new book | ⟨CHI⟩ asked for a new toy | ⟨CHI⟩ I love this ______ |
| **Image-Grounded Dialogue** | | ⟨Q⟩ can you tell what book it's reading ⟨A⟩ the marriage of true minds by stephen evans | | | what do we have here? ______ |

parameterized by $\boldsymbol{\theta}$, that the next word is $w$ conditioned on the context $c$. Here, $s_{\boldsymbol{\theta}}(w \mid c)$ quantifies the unexpectedness of predicting $w$, or the pointwise information carried by $w$ conditioned on the context.

The *grounding information gain* $G_{\boldsymbol{\theta}}(v)$ for $v$ is defined as

$$G_{\boldsymbol{\theta}}(v) = \frac{1}{N} \sum_{n=1}^{N} \left( \frac{1}{M} \sum_{u \neq v}^{M} \left[ s_{\boldsymbol{\theta}}\left(v_{⟨LAN⟩} \mid \overline{c}_n\left(u_{⟨ENV⟩}\right)\right) \right. \right.$$
$$\left. \left. - s_{\boldsymbol{\theta}}\left(v_{⟨LAN⟩} \mid \overline{c}_n\left(v_{⟨ENV⟩}\right)\right)\right] \right).$$

This is a sample-based estimation of the expected log-likelihood ratio between the match and mismatch conditions

$$G_{\boldsymbol{\theta}}(v) = \mathbb{E}_{c,u}\left[\log \frac{P_{\boldsymbol{\theta}}(v_{⟨LAN⟩} \mid c, v_{⟨ENV⟩})}{P_{\boldsymbol{\theta}}(v_{⟨LAN⟩} \mid c, u_{⟨ENV⟩})}\right],$$

which quantifies how much more information the matched ground provides for predicting the linguistic form, compared to a mismatched one. A positive $G_{\boldsymbol{\theta}}(v)$ indicates that the matched environmental token increases the predictability of its linguistic form. We report $G_{\boldsymbol{\theta}} = \frac{1}{|V|} \sum_{v \in V} G_{\boldsymbol{\theta}}(v)$, and track $G_{\boldsymbol{\theta}^{(t)}}$ across training steps $t$ to analyze how grounding emerges over time.

### 3.3. Model Training

We train LMs from random initialization, ensuring that no prior linguistic knowledge influences the results. Our training uses the standard causal language modeling objective, as in most generative LMs. To account for variability, we repeat all experiments with 5 random seeds, randomizing both model initialization and corpus shuffle order. Our primary architecture is Transformer (Vaswani et al., 2017) in the style of GPT-2 (Radford et al., 2019) with 18, 12, and 4 layers, with all of them having residual connections. We extend the experiments to 4-layer unidirectional LSTMs (Hochreiter & Schmidhuber, 1997) with no residual connections, as well as

12- and 4-layer state-space models (specifically, Mamba-2; Dao & Gu, 2024). For fair comparison with LSTMs, the 4-layer Mamba-2 models do not involve residual connections, whereas the 12-layer ones do. For multimodal settings, while standard LLaVA (Liu et al., 2023) uses a two-layer perceptron to project ViT embeddings into the language model, we bypass this projection in our case and directly feed the DINOv2 representations into the LM. We obtain the developmental trajectory of the model by saving checkpoints at various training steps, sampling more heavily from earlier steps, following Chang & Bergen (2022). Notice that datasets used to pretrain these LMs are the original version of those mentioned in section 3.1, and the context templates mentioned in section 3.2 are strictly inference-only and are never seen during model training.

## 4. Behavioral Evidence

### 4.1. Behavioral Evidence of Emergent Grounding

In this section, we ask: **Does symbol grounding emerge behaviorally in our settings?** We first test whether models show systematic surprisal reduction when predicting a linguistic token when its environmental counterpart is in context (Figure 2, where the gap between the lines represents the grounding information gain). For Transformers (Figures 2a and 2b) and Mamba-2 (Figure 2c), surprisal in the match condition decreases steadily while that in the mismatch condition enters a high-surprisal plateau early, indicating that the models leverage environmental context to predict the linguistic form. In contrast, the unidirectional LSTM (Figure 2d) shows little separation between the conditions, reflecting the absence of grounding. Overall, these results provide behavioral evidence of emergent grounding: in sufficiently expressive architectures (Transformers and Mamba-2), the correct environmental context reliably lowers surprisal for its linguistic counterpart, whereas LSTMs fail to exhibit this effect, marking an architectural boundary on where grounding can emerge.

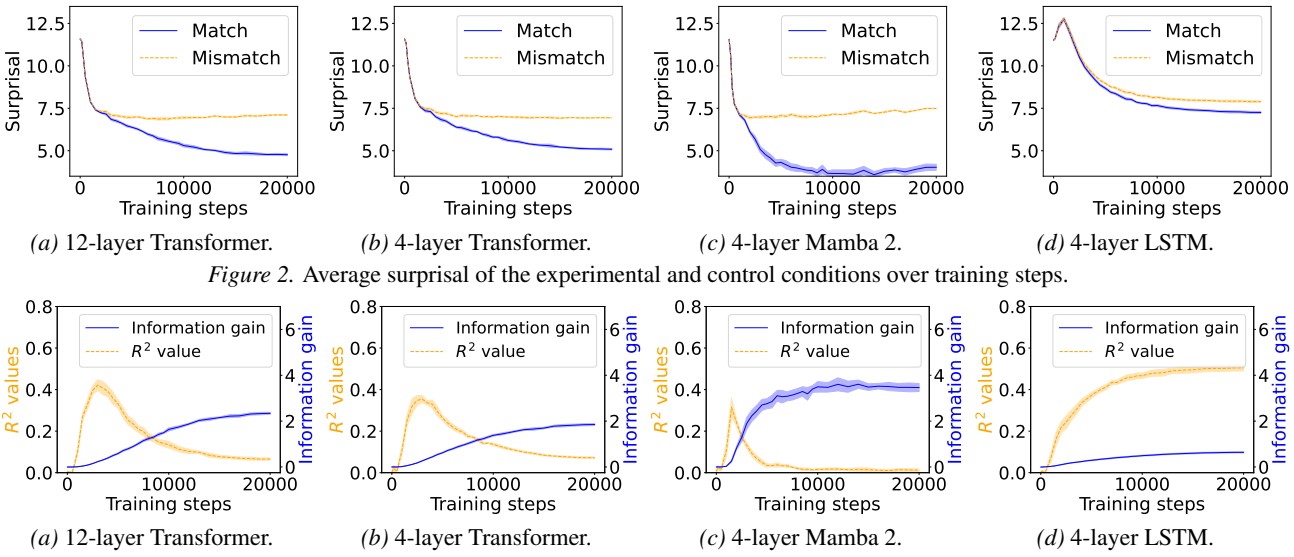

*Figure 2.* Average surprisal of the experimental and control conditions over training steps.

*(a)* 12-layer Transformer.  *(b)* 4-layer Transformer.  *(c)* 4-layer Mamba 2.  *(d)* 4-layer LSTM.

*Figure 3.* Grounding information gain and its correlation to the co-occurrence of linguistic and environment tokens over training steps.

## 4.2. Behavioral Effects Beyond Co-Occurrence

A natural concern is that the surprisal reductions might be fully explainable by shallow statistics: **the models might have simply memorized frequent co-occurrences of ⟨ENV⟩ and ⟨LAN⟩ tokens, without learning a deeper and more general mapping.** We test this hypothesis by comparing the tokens' co-occurrence with the grounding information gain in the child-directed speech data.

We define co-occurrence between the corresponding ⟨ENV⟩ and ⟨LAN⟩ tokens at the granularity of a 512-token training chunk. For each target word $v$, we count the number of chunks in which both its ⟨ENV⟩ and ⟨LAN⟩ tokens appear. Following standard corpus-analysis practice, these raw counts are log-transformed. For each model checkpoint, we run linear regression between the log co-occurrence and the grounding information gain of words, obtaining an $R^2$ statistic as a function of training time.

Figure 3 shows the $R^2$ values (orange) alongside the grounding information gain (blue) for different architectures. In both the Transformer and Mamba-2, $R^2$ rises sharply at the early steps but then goes down, even if the grounding information gain continues increasing. These results suggest that grounding in Transformers and Mamba-2 cannot be fully accounted for by co-occurrence statistics: while models initially exploit surface co-occurrence regularities, later improvements in grounding diverge from these statistics, indicating reliance on richer and more complicated features acquired during training. In contrast, LSTM shows persistently increasing $R^2$ but little increase in grounding information gain over training steps, suggesting that it encodes co-occurrence but lacks the architectural mechanism to transform it into predictive grounding.

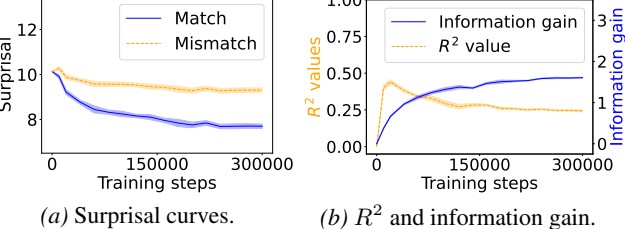

*(a)* Surprisal curves.  *(b)* $R^2$ and information gain.

*Figure 4.* Average surprisal of the experimental and control conditions, as well as the grounding information gain and its correlation to the co-occurrence of linguistic and environment tokens over training steps. All results are from a 12-layer Transformer model on image-grounded dialogue.

## 4.3. Results on Image-Grounded Dialogue

We next test whether the grounding effects observed in CHILDES generalize to VLMs, using the Visual Dialog dataset. In this setting, the environmental ground is supplied by image features (Table 1), and mismatched contexts are generated via image inpainting with Stable Diffusion 2 (Rombach et al., 2022), which regenerates the ground-truth mask region corresponding to the target referent.

We train 12-layer Transformers with 5 random seeds. Similarly as Figures 2a–2b and Figures 3a–3b, when images serve as the environmental ground, Transformers show a clear surprisal gap between match and mismatch conditions (Figure 4a), with the grounding information gain increasing steadily while $R^2$ peaks early and declines (Figure 4b), although the observed effect is slightly less pronounced. These results confirm that emergent grounding cannot be fully explained by co-occurrence statistics.

Overall, our findings demonstrate that Transformers are able to exploit environmental grounds in various modalities to facilitate linguistic prediction. The smaller but consistent

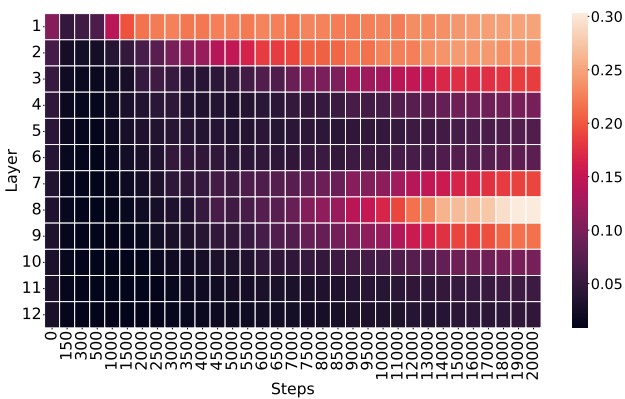

*(a)* Saliency of layer-wise attention from environmental to linguistic tokens across training steps.

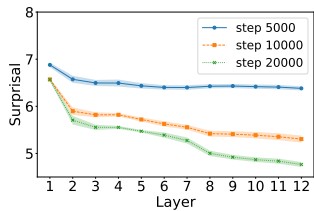

*(b)* Layer-wise tuned lens surprisals in the matched condition.

*Figure 5.* Mechanistic analysis of symbol grounding emergence.

gains in the image-grounded case suggest that while grounding from visual tokens is harder, the same architectural dynamics identified in textual testbeds still apply.

# 5. Mechanistic Explanation

In this section, we provide a mechanistic and interpretable account of the previous observation. We first draw hypotheses from a 12-layer Transformer trained on CHILDES with 5 random seeds, and extend the experiments to image-grounded dialogue (Section 5.4).

## 5.1. The Emergence of Symbol Grounding

To provide a mechanistic account of symbol grounding, i.e., when it emerges during training and how it is represented in the network, we apply two interpretability analyses.

**Saliency flow.** For each layer $\ell$, we compute a saliency matrix following Wang et al. (2023): $I_\ell = \left| \sum_h A_{h,\ell} \odot \frac{\partial \mathcal{L}}{\partial A_{h,\ell}} \right|$, where $A_{h,\ell}$ denotes the attention matrix of head $h$ in layer $\ell$. Each entry of $I_\ell$ quantifies the contribution of the corresponding attention weight to the cross-entropy loss $\mathcal{L}$, averaged across heads. Our analysis focuses on ground-to-symbol connections, i.e., flows from environmental ground ($\langle\text{ENV}\rangle$) tokens to the token immediately preceding (and predicting) their linguistic forms ($\langle\text{LAN}\rangle$).

**Probing with the Tuned Lens.** We probe layer-wise representations using the Tuned Lens (Belrose et al., 2023), which trains affine projectors to map intermediate hidden

*Table 2.* Causal intervention results on identified gather and aggregate heads across training checkpoints (ckpt.), with a threshold value of 30% as defined in section 5.3. **Avg. Count** denotes the average number of heads of each type over inference times, and **Avg. Layer** denotes the average layer index where they appear. **Interv. Sps.** reports surprisal after zeroing out the identified heads, while **Ctrl. Sps.** reports surprisal after zeroing out an equal number of randomly selected heads. **Original** refers to the baseline surprisal without any intervention. *** indicates a significant result ($p < 0.001$) where the intervention surprisal is higher than that in the corresponding control experiment.

| Ckpt. | Gather Head | | | | Aggregate Head | | | | Original |
|---|---|---|---|---|---|---|---|---|---|
| | Avg. Count | Avg. Layer | Interv. Sps. | Ctrl. Sps. | Avg. Count | Avg. Layer | Interv. Sps. | Ctrl. Sps. | |
| 500 | 0.00 | - | - | - | 0.07 | 8.74 | 9.34 | 9.34 | 9.34 |
| 5k | 0.35 | 3.32 | 6.37 | 6.38 | 2.28 | 7.38 | **6.51** (***) | 6.39 | 6.38 |
| 10k | 3.26 | 3.67 | 5.25 | 5.32 | 5.09 | 7.28 | **5.86** (***) | 5.29 | 5.30 |
| 20k | 5.76 | 3.59 | 4.69 | 4.79 | 6.71 | 7.52 | **5.62** (***) | 4.76 | 4.77 |

states to the final prediction space while keeping the LM output head frozen.

**Results.** Ground-to-symbol saliency is weak in the early stages of training but rises sharply later, peaking in layers 7–9 (Figure 5a), suggesting that mid-layer attention may play a central role in establishing symbol–ground correspondences. In addition, Figure 5b shows that early layers remain poor predictors even at late training stages (e.g., after 20,000 steps), whereas surprisal begins to drop markedly from layer 7 at intermediate stages (step 10,000), suggesting a potential representational shift in the middle layers.

## 5.2. Hypothesis: Gather-and-Aggregate Heads Implement Symbol Grounding

Building on these results, we hypothesize that specific Transformer heads in the middle layers enable symbol grounding. To test the hypothesis, we examine attention saliencies for selected heads (Figure 6). We find that several heads exhibit patterns consistent with the gather and aggregate mechanisms described by Bick et al. (2025): gather heads (e.g., Figures 6a and 6b) compress relevant information into a subset of positions, while aggregate heads (e.g., Figures 6c and 6d) redistribute this information to downstream tokens. In our setups, saliency often concentrates on environmental tokens such as $train_{\langle\text{ENV}\rangle}$, where gather heads pool contextual information into compact, retrievable states. In turn, aggregate heads broadcast this information from environmental ground ($train\langle\text{ENV}\rangle$) to the token immediately preceding the linguistic form, thereby supporting the prediction of $train_{\langle\text{LAN}\rangle}$. Taking these observations together, we hypothesize that the gather-and-aggregate heads implement the symbol grounding mechanism.

## 5.3. Causal Interventions of Attention Heads

We then conduct causal interventions of attention heads to validate our previous hypothesis.

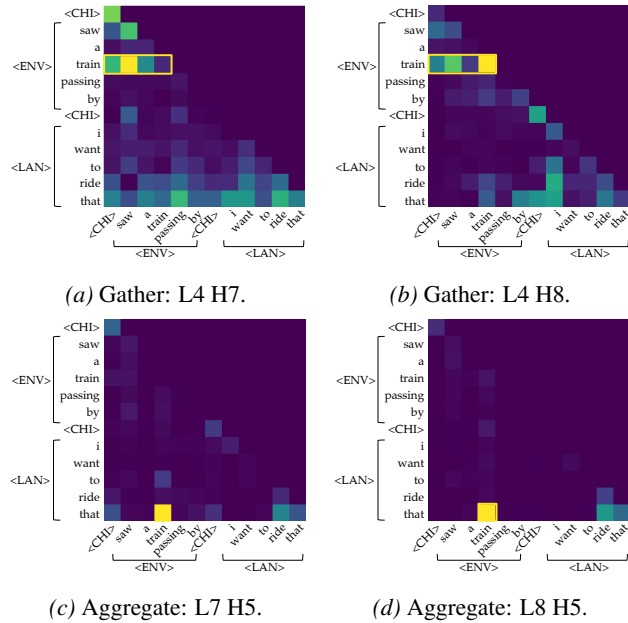

*(a)* Gather: L4 H7.   *(b)* Gather: L4 H8.

*(c)* Aggregate: L7 H5.   *(d)* Aggregate: L8 H5.

*Figure 6.* Examples of gather and aggregate heads identified. L: layer; H: head.

**Operational definition.** We identify attention heads as gather or aggregate following these standards:

- **Gather head.** An attention head is classified as a gather head if at least 30% of its total saliency is directed toward the environmental ground token from the previous ones.

- **Aggregate head**: An attention head is classified as an aggregate head if at least 30% of its total saliency flows from the environmental ground token to the token immediately preceding the corresponding linguistic token.

**Causal intervention methods.** In each context, we apply causal interventions to the identified head types and their corresponding controls. Following Bick et al. (2025), interventions are implemented by zeroing out the outputs of heads. For the control, we mask an equal number of randomly selected heads in each layer, ensuring they do not overlap with the identified gather or aggregate heads.

**Results and discussions.** As training progresses, the number of both gather and aggregate heads increases (Table 2), suggesting that these mechanisms emerge over the course of learning. Causal interventions reveal a clear dissociation: zeroing out aggregate heads consistently produces significantly higher surprisal compared to controls, whereas the gather head interventions have no such effect. This asymmetry suggests that gather heads serve in a role less critical in our settings, where the input template is semantically light and the environmental evidence alone suffices to shape the linguistic form. Layer-wise patterns further support this division of labor: gather heads cluster in shallow layers (3-4), while aggregate heads concentrate in mid layers (7-8). This resonates with our earlier probing results, where surprisal re-

| Thres. | Ckpt. | Aggregate Head | | | | Original |
|---|---|---|---|---|---|---|
| | | Avg. Count | Avg. Layer | Interv. Sps. | Ctrl. Sps. | |
| 70% | 20k | 32.30 | 7.78 | 9.96 | 9.95 | 9.21 |
| | 100k | 35.63 | 7.71 | **9.42** (***) | 8.84 | 8.24 |
| | 200k | 34.99 | 7.80 | **8.95** (***) | 8.15 | 7.76 |
| | 300k | 34.15 | 7.76 | **8.96** (***) | 8.11 | 7.69 |
| 90% | 20k | 10.66 | 8.33 | **9.51** (***) | 9.43 | 9.21 |
| | 100k | 13.90 | 8.26 | **8.95** (***) | 8.50 | 8.24 |
| | 200k | 13.47 | 8.46 | **8.41** (***) | 7.88 | 7.76 |
| | 300k | 12.73 | 8.42 | **8.40** (***) | 7.87 | 7.69 |

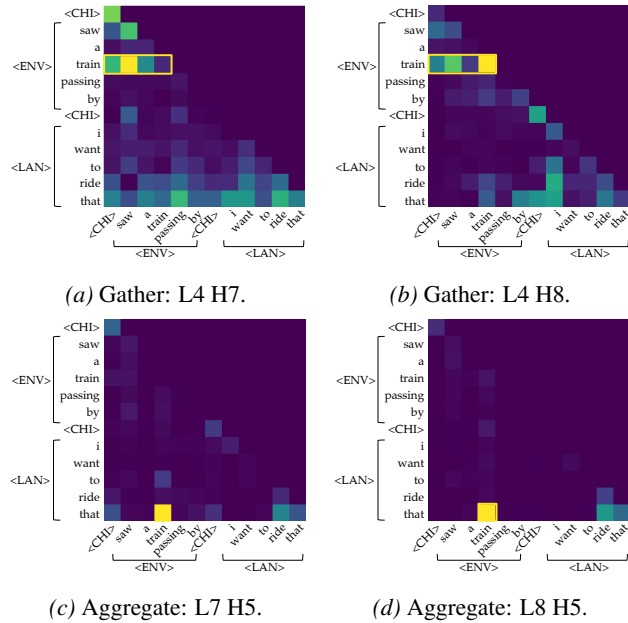

*Figure 7.* Mechanistic analysis in the image-grounded visual dialogue setting. Top: Causal intervention results on identified aggregate heads across training checkpoints, where intervention on aggregate heads consistently yields significantly higher surprisal ($p < 0.001$, ***) compared to the control group ones. Bottom: Saliency of layer-wise attention from environmental tokens (i.e., image tokens corresponding to patches within the bounding boxes of the target object) to linguistic tokens across training steps.

ductions became prominent only from layers 7-9. Together, these findings highlight aggregate heads in the middle layers as the primary account of grounding in the model.

### 5.4. Generalization to Visual Dialog with Images

We also conduct causal interventions on the VLM's attention heads to further validate our hypothesis above.

**Operational definition.** We define an attention head as an aggregate head if at least a certain portion (70% or 90% in our experiments) of its total image-to-text saliency flows from patches within the bounding box to the token immediately preceding the corresponding linguistic token.

**Causal intervention methods.** Similarly to Section 5.3, we apply causal interventions to the identified aggregate heads and their corresponding controls, by zeroing out their outputs. For the control, we mask an equal number of randomly selected heads in each layer, ensuring they do not overlap with the identified aggregate heads.

**Results and discussions.** As training progresses, the number of aggregate heads increases first and then becomes steady (Figure 7, top), suggesting that these mechanisms emerge over the course of learning. Zeroing out aggregate heads consistently produces significantly higher surprisal compared to the controls. The average layer depth of these heads also aligns well with the saliency heatmap (Figure 7, bottom).

## 6. Discussions

**Generalization to full-scale VLMs.** As an additional case study, we extend our grounding-as-aggregation hypothesis to a full-scale VLM, LLaVA-1.5-7B (Liu et al., 2023). Even in this heavily engineered architecture, we identify many attention heads exhibiting aggregation behavior consistent with our earlier findings (Figure 1b), reinforcing the view that symbol grounding arises from specialized heads. Meanwhile, full-scale models like LLaVA introduce additional complications: they incorporate CLIP-derived embeddings that already encode language priors for better performance, and global information may be stored in tokens out of the object regions (Darcet et al., 2024). Moreover, the large number of visual tokens substantially increases both computational cost and the difficulty of isolating genuine aggregation heads. These factors make systematic identification and intervention at scale a nontrivial challenge. For these reasons, while our case study highlights promising evidence of grounding heads in modern VLMs, systematic detection and causal evaluation of such heads at scale remains an open challenge. Future work will need to develop computationally viable methods for detecting aggregation heads and applying causal interventions to validate their roles.

**Connection to philosophical conceptualizations.** While the primary purpose of our minimal testbed (child-direct speech setting; Table 1) is to offer a proxy for understanding grounding, experiments in this setting can be naturally viewed as investigations of symbol binding in language models, the problem that studies how symbols are connected together. This work extends the activation-based study on symbol binding (Feng & Steinhardt, 2024; Dai et al., 2024; Feng et al., 2025) and offers evidence on the attention-head level, showing that aggregate heads are crucial in implementing the mechanism (Table 2). In line with Yang et al. (2025), our work suggests that attention heads are crucial for implementing symbolic structures in LMs, and provides more controlled and causal evidence.

**The Philosophical Roots of Grounding.** Our findings highlight the need to sharpen the meaning of grounding in multimodal models. Prior work has often equated grounding with statistical correlations between visual and textual signals, such as attention overlaps or geometric alignments (Cao et al., 2024; Bousselham et al., 2024; Schnaus et al., 2025).

While informative, such correlations diverge from the classic formulation by Harnad (1990), which requires symbols to be causally anchored to their referents in the environment. In line with Harnad (1990), we frame grounding as a mechanistic property: one that can be traced along training, observed in the specialization of attention heads, and validated through causal interventions, providing a protocol for diagnosing when and how models genuinely tie symbols to meaning rather than mere correlations. On another line, our results, which show that aggregate heads implement symbol grounding (and binding), echo Pavlick (2023) in arguing that *LLMs lack the capacity to represent abstract symbolic structure* should not be accepted a priori. Instead, such claims should be evaluated carefully and empirically, with the focus on uncovering the models' underlying competence rather than drawing conclusions solely from high-level architectures and surface-level performance.

**Practical implications to LM hallucinations.** Our findings have practical implications for improving the reliability of LM outputs: by identifying aggregation heads that mediate grounding between environmental and linguistic tokens, we provide a promising mechanism to detect model reliability before generation. Our findings echo a pathway to mitigate hallucinations by focusing on attention control: many hallucination errors stem from misallocated attention in intermediate layers (Jiang et al., 2025; Chen et al., 2024b). Such attention-level signals can serve as early indicators of overtrust or false grounding, motivating practical solutions like decoding-time strategies to mitigate and eventually prevent hallucination (Huang et al., 2024).

## Acknowledgements

This work was supported in part by NSF IIS-1949634, NSF SES-2128623, NSERC RGPIN-2024-04395, the Weinberg Cognitive Science Fellowship to Ziqiao Ma, a Vector Scholarship to Xiaoxi Luo, and a Canada CIFAR AI Chair award to Freda Shi. The authors would like to thank Songlin Yang and Jing Ding for their valuable feedback.

## Impact Statement

This study analyzes the mechanistic emergence of symbol grounding in (multimodal) language models using publicly available datasets such as CHILDES and Visual Dialog, and offers an example of pipelining interpretability techniques to understand large-scale neural networks. All the images are from the publicly available MSCOCO dataset (Lin et al., 2014), where no personally identifiable data is used.

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

# A. Dataset Details

## A.1. Context Templates (Inference Time)

We select the target tokens following the given procedure:

1. Get a list of words, with their ENV and LAN frequency both greater than or equal to 100 in the CHILDES dataset;

2. Get another list of nouns from CDI;

3. Take intersection and select top 100 words (by frequency of their ENV token) as target token list.

In CHILDES, all contexts are created with `gpt-4o-mini` followed by human verification if the genrated contexts are semantically light. Notice that these contexts are only used during inference. We adopt the following prompt:

---

**Prompt Templates for CHILDES**

```
Given the word "{word}", create 3 pairs of sentences that follow this
requirement:
1. The first sentence has a subject "The child", describing an event or
situation, and has the word "{word}". Make sure to add a newline to the end of
this first sentence
2. The second sentence is said by the child (only include the speech itself,
don't include "the child say", etc.), and the word "{word}" also appears in
the sentence said by the child. Do not add quote marks either
3. Print each sentence on one line. Do not include anything else.
4. Each sentence should be short, less than 10 words.
5. The word "{word}" in both sentence have the same meaning and have a clear
indication or an implication relationship.
6. "{word}" should not appear at the first/second word of each sentence.
Generate 3 pairs of such sentences, so there should be 6 lines in total.
You should not add a number.
For each line, just print out the sentence.
```

---

In visual dialogue (caption version and VLM version), we pre-define 10 sets of templates for each version:

---

**Prompt Templates for Visual Dialogue (Caption Version)**

```
this:<ENV> is:<ENV> [FILLER]:<ENV> <Q> what:<LAN> is:<LAN> it:<LAN> <A>
(predict [FILLER]:<LAN>)

this:<ENV> is:<ENV> [FILLER]:<ENV> <Q> what:<LAN> do:<LAN> you:<LAN>
call:<LAN> this:<LAN> <A> (predict [FILLER]:<LAN>)

this:<ENV> is:<ENV> [FILLER]:<ENV> <Q> can:<LAN> you:<LAN>
name:<LAN> this:<LAN> object:<LAN> <A>
(predict [FILLER]:<LAN>)

this:<ENV> is:<ENV> [FILLER]:<ENV> <Q> what's:<LAN>
this:<LAN> called:<LAN> <A>
(predict [FILLER]:<LAN>)
```

---

**Prompt Templates for Visual Dialogue (Caption Version) (continued)**

```
this:<ENV> is:<ENV> [FILLER]:<ENV> <Q> what:<LAN>
this:<LAN> thing:<LAN> is:<LAN> <A>
(predict [FILLER]:<LAN>)

this:<ENV> is:<ENV> [FILLER]:<ENV> <Q> what:<LAN>
would:<LAN> you:<LAN> name:<LAN> this:<LAN> <A>
(predict [FILLER]:<LAN>)

this:<ENV> is:<ENV> [FILLER]:<ENV> <Q>
what's:<LAN> the:<LAN> name:<LAN> of:<LAN> this:<LAN>
item:<LAN> <A> (predict [FILLER]:<LAN>)

this:<ENV> is:<ENV> [FILLER]:<ENV> <Q> how:<LAN>
do:<LAN> you:<LAN> identify:<LAN> this:<LAN> <A>
(predict [FILLER]:<LAN>)

this:<ENV> is:<ENV> [FILLER]:<ENV> <Q> what:<LAN>
do:<LAN> we:<LAN> have:<LAN> here:<LAN> <A>
(predict [FILLER]:<LAN>)

this:<ENV> is:<ENV> [FILLER]:<ENV> <Q> how:<LAN>
do:<LAN> you:<LAN> call:<LAN> this:<LAN>
object:<LAN> <A> (predict [FILLER]:<LAN>)
```

**Prompt Templates for Visual Dialogue (VLM Version)**

```
"<image> \nwhat is it ?",
"<image> \nwhat do you call this ?",
"<image> \ncan you name this object ?",
"<image> \nwhat is this called ?",
"<image> \nwhat this thing is ?",
"<image> \nwhat would you name this ?",
"<image> \nwhat is the name of this item ?",
"<image> \nhow do you identify this ?",
"<image> \nwhat do we have here ?",
"<image> \nhow do you call this object ?"
```

### A.2. Word List for CHILDES and Vision Dialogue (Text Only)

[box, book, ball, hand, paper, table, toy, head, car, chair, room, picture, doll, cup, towel, door, mouth, camera, duck, face, truck, bottle, puzzle, bird, tape, finger, bucket, block, stick, elephant, hat, bed, arm, dog, kitchen, spoon, hair, blanket, horse, tray, train, cow, foot, couch, necklace, cookie, plate, telephone, window, brush, ear, pig, purse, hammer, cat, shoulder, garage, button, monkey, pencil, shoe, drawer, leg, bear, milk, egg, bowl, juice, ladder, basket, coffee, bus, food, apple, bench, sheep, airplane, comb, bread, eye, animal, knee, shirt, cracker, glass, light, game, cheese, sofa, giraffe, turtle, stove, clock, star, refrigerator, banana, napkin, bunny, farm, money]

*Table 3.* Training and test examples across datasets with target word *book*. The training examples combine environmental tokens ( ⟨ENV⟩; shaded ) with linguistic tokens (⟨LAN⟩). Test examples are constructed with either matched (*book*) or mismatched (*toy*) environmental contexts, paired with corresponding linguistic prompts. Note that in caption-grounded dialogue, $book_{⟨ENV⟩}$ and $book_{⟨LAN⟩}$ are two distinct tokens received by LMs.

| Dataset | Training Example | | Test Example | | |
|---|---|---|---|---|---|
| | ⟨ENV⟩ | ⟨LAN⟩ | ⟨ENV⟩ **Match** | ⟨ENV⟩ **Mismatch** | ⟨LAN⟩ |
| **Caption-Grounded Dialogue** | *a dog appears to be reading a book with a full bookshelf behind* | *⟨Q⟩ can you tell what book it's reading ⟨A⟩ the marriage of true minds by stephen evans* | *this is a book* | *this is a toy* | *⟨Q⟩ can you name this object ⟨A⟩* ————— |

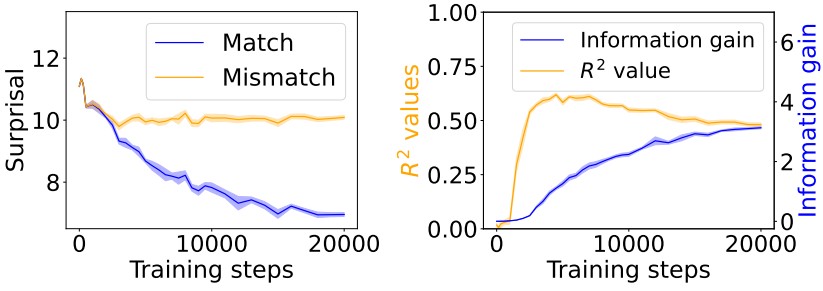

*(a)* Surprisal curves (w/ caption).     *(b)* $R^2$ and information gain (w/ caption).

*Figure 8.* Average surprisal of the experimental and control conditions, as well as the grounding information gain and its correlation to the co-occurrence of linguistic and environment tokens over training steps. All results are from a 12-layer Transformer model on grounded dialogue data.

### A.3. Word List for Vision Dialogue (VLM)

[box, book, table, toy, car, chair, doll, door, camera, duck, truck, bottle, bird, elephant, hat, bed, dog, spoon, horse, train, couch, necklace, cookie, plate, telephone, window, pig, cat, monkey, drawer, bear, milk, egg, bowl, juice, ladder, bus, food, apple, sheep, bread, animal, shirt, cheese, giraffe, clock, refrigerator, accordion, aircraft, alpaca, ambulance, ant, antelope, backpack, bagel, balloon, barrel, bathtub, beard, bee, beer, beetle, bicycle, bidet, billboard, boat, bookcase, boot, boy, broccoli, building, bull, burrito, bust, butterfly, cabbage, cabinetry, cake, camel, canary, candle, candy, cannon, canoe, carrot, cart, castle, caterpillar, cattle, cello, cheetah, chicken, chopsticks, closet, clothing, coat, cocktail, coffeemaker, coin, cosmetics]

## B. Implementation Details

### B.1. Checkpointing

#### B.1.1. CHILDES, VISUAL DIALOGUE WITH CAPTIONS

We save the intermediate steps: [0, 150, 300, 500, 1000, 1500, 2000, 2500, 3000, 3500, 4000, 4500, 5000, 5500, 6000, 6500, 7000, 7500, 8000, 8500, 9000, 9500, 10000, 11000, 12000, 13000, 14000, 15000, 16000, 17000, 18000, 19000, 20000] (33 checkpoints in total)

#### B.1.2. VISUAL DIALOGUE (VLM)

We save the intermediate steps: [10000, 20000, 40000, 60000, 80000, 100000, 120000, 140000, 160000, 180000, 200000, 220000, 240000, 260000, 280000, 300000] (16 checkpoints in total)

## C. Addendum to Results

### C.1. Caption-grounded dialogue

We use the same dataset as the image-grounded dialogue, the Visual Dialog dataset (Das et al., 2017). In our setup, MSCOCO captions serve as the environmental tokens (⟨ENV⟩) and the dialogue turns form the linguistic tokens (⟨LAN⟩). In this pseudo

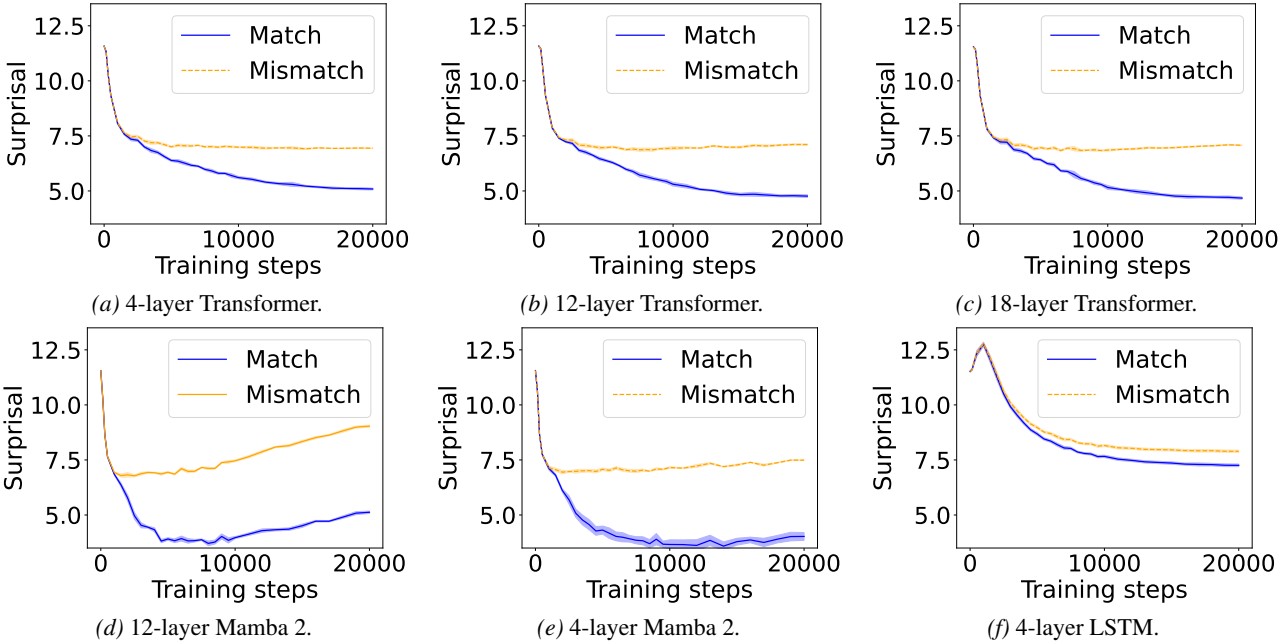

*Figure 9.* Average surprisal of the experimental and control conditions over training steps.

cross-modal setting, textual descriptions of visual scenes ground natural conversational interaction. Compared to CHILDES, this setup introduces richer semantics and longer utterances, while still using text-based inputs for both token types, thereby offering a stepping stone toward grounding in fully visual contexts.

To assess symbol grounding in caption-grounded dialogue, we use a similar contrastive test as Child-directed speech and Image-grounded Dialogue, as demonstrated in Table 1.

In this setting, we also train 12-layer Transformers with 5 random seeds. Compared to Figures 4a–4b, there is a similar but stronger pattern: a larger surprisal gap exists between match and mismatch conditions (Figure 8a), with the grounding information gain increasing steadily while $R^2$ peaks early and declines (Figure 8b). Both settings confirm that emergent grounding cannot be fully explained by co-occurrence statistics.

### C.2. Detailed Behavioral Analysis for all Models

We show the complete behavioral evidence for all models in Figure 9, and co-occurrence analysis in Figure 10. On top of that, for 12-layer Transformers, a beeswarm plot indicating per context match/mismatch surprisal is shown in Figure 11.

### C.3. Detailed Gather and Aggregate Analysis (Transformer)

After finding the set of gather and aggregate heads for each context, we run an overtime analysis showing the proportion of saliency to the total saliency, as is shown in Figure 12.

### C.4. Relationship between Aggregate Head Number and Grounding Information Gain

We examine the 12-layer Transformer LM for both the child-directed speech and VLM settings. For each target token (detailed in Sections A.2 and A.3), we compare its grounding information gain with the average aggregate head it has (Figure 13). We find an intermediate-to-strong positive correlation between the grounding information gain and the aggregate head number detected.

## D. LLM Statement

In this work, large language models (LLMs) are employed in two limited ways: (i) to polish the writing and improve the linguistic clarity of the paper; (ii) to assist in code writing and debugging. LLMs are not involved in the design of the core method, the experimental setup, data analysis, or the interpretation of the results. All texts presented in the paper, as well as

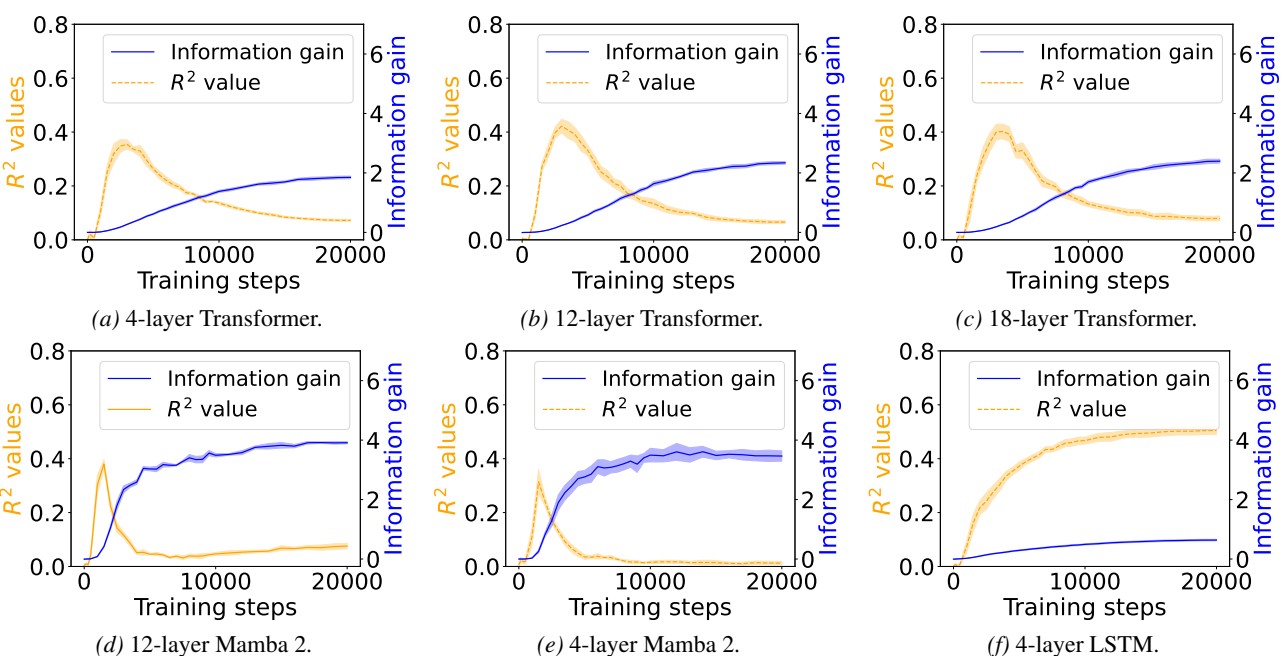

*Figure 10.* Grounding information gain and its correlation to the co-occurrence of linguistic and environment tokens over training steps.

the code, are endorsed by the authors, and the authors take full responsibility of the content presented in this paper.

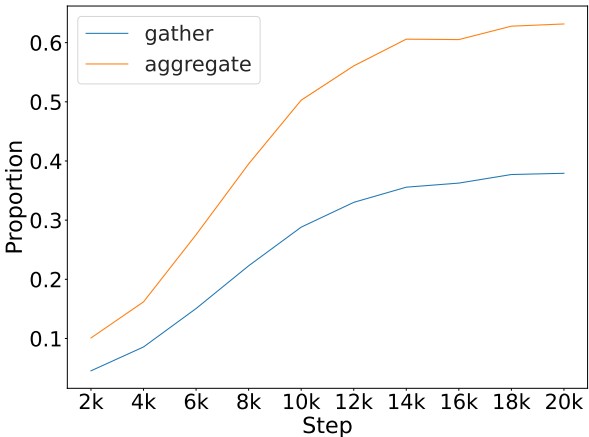

*Figure 12.* Gather-and-aggregate overtime strength.

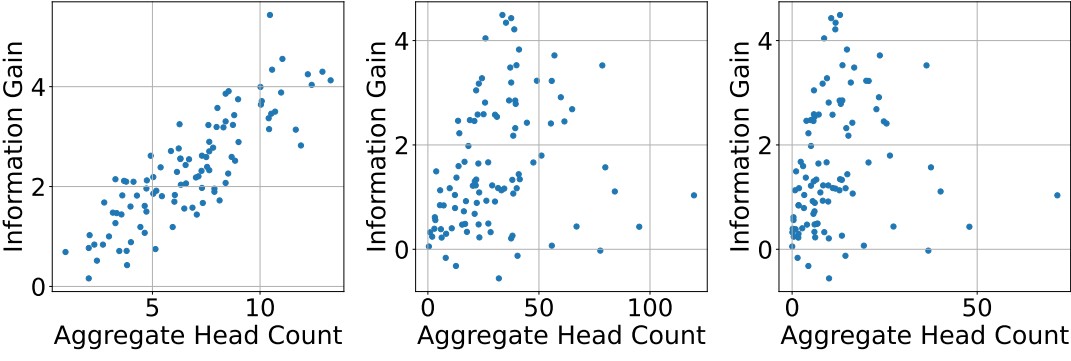

*(a)* Information Gain vs Aggregate Head Number (CHILDES), $R^2 = 0.847$.

*(b)* Information Gain vs. Aggregate Head Number (VLM, with 70% threshold), $R^2 = 0.250$.

*(c)* Information Gain vs. Aggregate Head Number (VLM, with 90% threshold), $R^2 = 0.175$.

*Figure 13.* Grounding information gain vs. number of aggregate heads.

