# OpenReview forum: "The Mechanistic Emergence of Symbol Grounding in Language Models"
_ICML.cc/2026/Conference — ICML 2026 regular_

### Official Review · Reviewer_T9jK · 2026-02-17

**Soundness:** 3
**Presentation:** 3
**Significance:** 3
**Originality:** 4
**Overall Recommendation:** 5
**Confidence:** 4

**Summary:**

The paper studies "symbol grounding" in vision-language models (VLMs) by considering a controlled setup in which models are tested across training for the extent to which the presence of an object in a scene makes them more or less likely to predict the verbal name of the object. The paper further attributes the observed emergence of grounding to specific "aggregate" heads and extends the results to a number of architectures and settings.

**Compliance With Llm Reviewing Policy:**

Affirmed.

**Final Justification:**

This is an interesting and clear paper dealing with an interesting topic. The rebuttal clarified several questions, and I think the paper is ready to be presented at ICML.

**Key Questions For Authors:**

1) Can you explain in more detail how you constructed the training corpora. Besides what I ask in the box above, can you also explain the rationale for sampling words according to two criteria (1 and 2 in A.1).

2) Is grounding different from any other kind of context-based token-to-token learning (see comments in the box above)?

3) It seems that figures 1a and 1c are not referenced in the text, and figure 1b is only mentioned towards the end. I found this confusing. Also, I find the picture in 1b too dark.

4a) You measured the information gain for an object name when the object is in context vs when it isn't. How about a comparison between the surprisal for a grounded word vs an ungrounded word, with stable environment, particularly for the image-grounded setup. I thought that this could be a control related to the following point...

4b) It would be interesting to see how the mismatched-contexts images generated with Stable Diffusion look like: how natural are they? There is an example in Table 1, but it's tiny. I just wonder about how natural they are...

5) In 5.1, you report that early layers are poor predictors even at late stages of training (fig 5b). However, fig 5a would suggest that ground-to-symbol saliency is relatively high for early layers as well.

6) Too bad fig. 6 is in an appendix: perhaps, it could in part replace the currently largely unreferenced fig 1?

**Limitations:**

Yes, although it would be perhaps good to have a separate Limitations section, rather than hiding the limitations in the Discussion(s).

**Strengths And Weaknesses:**

Strengths

This is an interesting paper dealing with a core topic in multimodal AI, with a clean experimental paradigm and presenting some well-described findings.

Weaknesses

I was confused about the corpora used to train the models for the CHILDES and Visual Dialogue experiments. In particular, if I understood appendix A correctly, the data are not directly extracted CHILDES corpus, but generated by a language model using words from CHILDES? That seems to be a very different setup from what I'd expect a CHILDES-based corpus to consist in. Also, as it seems to specifically focus on environments that prominently feature an object, and sentences that mention that object, I wonder if this setup can really be claimed to lack explicit grounding supervision. This "hidden supervision" effect seems to be even more marked in the Visual Dialogue templates.

I do not know if this is a weakness, but I wonder to what extent the paper demonstrates the emergence of "grounding" as something different from the emergence of a dependency between two tokens (of which one happens to be a visual token). Is the claim then that grounding is exactly that--learning the relation between tokens that happen to be in different modalities, but through learning mechanisms that could also apply to the same modality? In this respect, the experiment looking at the relationship between co-occurrence and grounding information gain was interesting. It reminded me of what people did in the old Latent-Semantic-Analysis days, where one could show that similar contexts could make word representations similar even when the words did not co-occur (and even if the contexts themselves were not overlaping). Is what we are observing here essentially the same?

I appreciated that the related work included pre-deep-learning era studies. A further early work whose corpus seems quite similar to the one you use is the following: https://aclanthology.org/N16-1043/

I have other comments, but they are minor, and I will present them below as "Key Questions For Authors".

---

> ### Author Rebuttal · Authors · 2026-03-31
>
> We thank the reviewer for their thoughtful and constructive feedback and are encouraged by their positive assessment of our work.
>
> ## Weakness 1: Pretraining Data and "Hidden Supervision" (paragraph 1)
>
> We clarify that our pretraining corpus consists of the original, naturalistic CHILDES (22M tokens) and Visual Dialog (25M tokens) dataset (not generated by a language model), which are used to pretrain a GPT-2 model from scratch.
>
> The structured formats in the Appendices are inference-only evaluation templates used to measure grounding ability; they are never present in the pretraining corpora. The “100 nouns” refers specifically to the subset of concrete nouns we selected for evaluation (selection criteria described in Appendix A.1). Each noun receives 10 templates, yielding 1,000 test instances in total.
>
> Regarding "hidden supervision", our training data is noisy and lacks word-level alignment. In both CHILDES and Visual Dialog, an environment or image often contains multiple objects (or none at all), with no guarantee that a specific object will be mentioned in the text. There is no explicit signal indicating which word grounds to which object during training.
>
> ## Weakness 2 & Question 2: Definition about grounding, its learning mechanism, and comparison with LSA (paragraph 2)
> > I wonder to what extent the paper demonstrates the emergence of 'grounding' as something different from the emergence of a dependency between two tokens
>
> We would like to emphasize the fundamental difference between linguistic tokens (symbolic) and "visual tokens" (non-symbolic). In short, "visual tokens" in VLMs are continuously valued vectors produced by a pretrained neural image encoder, rather than discrete symbolic token. We kindly refer the reviewer to the response to Reviewer B8qm "Weakness 1 & Weakness 3" for details. Therefore, our work extends beyond learning the "dependency between two tokens", or "context-based token-to-token learning" (also Question 2).
> >  Is the claim then that grounding is exactly that ... could also apply to the same modality?
>
> Although it needs to be further validated, we believe that the mechanism of grounding, especially the aggregate head, is general, regardless of the modalities involved.
>
> Regarding LSA, we appreciate these interesting questions. Yet, as previously discussed, our conclusions go beyond LSA for two reasons:
> 1. Unlike LSA, which relies on discrete symbol co-occurrence, our VLM contexts consist of continuous, non-symbolic visual tensors.
> 2. Even in our CHILDES testbed, the learned relationship between `<ENV>` and `<LAN>` tokens is non-trivial from a distributional semantics perspective. `<ENV>` and `<LAN>` tokens appear in distinct contexts--environmental descriptions versus child-directed speech. From the LSA perspective, an `<ENV>` token is often more similar to other environmental tokens than to its linguistic counterpart, suggesting that grounding is not reducible to simple context overlap.
>
> ## Question 1
>
> For training corpora, please refer to the response to "Weakness 1".
> For word sampling criteria, please refer to the response to Reviewer bdw5 "Question 1" for details.
>
> ## Question 3&6
> Thank you for the suggestion. We will explicitly refer to Figures 1a and 1c to improve the clarity of our presentation, and move Figure 6 to the main text. While ICML does not allow re-uploading pdf during rebuttal, we will make these changes in the camera-ready version.
>
> ## Question 4
> a) We appreciate this thoughtful proposal. While we previously considered comparing surprisal of grounded vs. ungrounded words within a fixed context, we found it an unreliable control due to intrinsic model biases, such as word frequency. For example, in our  GPT-2-based VLM, the token "cat" may naturally yield lower surprisal than "cheetah" simply because it is a far more frequent word in the pretraining corpus, even when a cheetah is visually present. By keeping the target token fixed and only varying the environmental condition, we effectively isolate the grounding effect from these confounders.
>
> b) We have manually sampled many images and confirmed their natural appearance, which is why we utilized inpainting rather than simply blacking out bounding boxes. We provide a pair of original and inpainted (control) images:
> - original: https://anonymous.4open.science/r/emg-rebuttal-B8BE/book-test.jpg
> - inpainted: https://anonymous.4open.science/r/emg-rebuttal-B8BE/book-test-control.jpg
>
> ## Question 5
> Saliency flow is generally stronger in early layers across all token interactions. This image (https://anonymous.4open.science/r/emg-rebuttal-B8BE/context_normal_saliency_over_time.png) shows three types of saliency flows ("anchor2end" is the environmental to linguistic token saliency, as in Fig 5(a)), and in early layers these flows are consistently stronger.
>
> Also, as few aggregate heads are found in early layers, the limited prediction ability in early layers is not surprising.

---

> > ### Author Rebuttal · Reviewer_T9jK · 2026-03-31
> >
> > I thank the authors for their clarification. The ones about the pre-training corpus and the nature of visual representations should really be taken into account when revising the paper, as they were confusing to me despite an attentive reading.
> >
> > I find it a bit disappointing that the authors replied in a defensive way, as if I was accusing them of not innovating with respect to LSA (a research line from 30 years ago), when I was simply making some positive remarks on the mechanisms uncovered in the paper.

---

> > > ### Author Response · Authors · 2026-04-07
> > >
> > > Thank you again for the kind note! We apologize if our rebuttal came across as defensive, which was certainly not our intention and was partly a byproduct of condensing our response to meet the 5,000-character limit. We greatly appreciate the reviewer's valuable feedback and thoughtful comments, and we will incorporate them to further clarify and strengthen the paper.

---

### Official Review · Reviewer_UYs7 · 2026-03-10

**Soundness:** 2
**Presentation:** 3
**Significance:** 2
**Originality:** 3
**Overall Recommendation:** 3
**Confidence:** 4

**Summary:**

This paper investigates where and how symbol grounding emerges in large language models (LLMs) trained with next-token prediction. Using a controlled grounded-language setup and a surprisal-based metric, the authors show that grounding-like effects emerge in Transformers and state-space models like Mamba, but not in unidirectional LSTMs, and are not fully explained by co-occurrence statistics. Through saliency analyses, tuned-lens probing, and causal interventions on attention heads, the paper further argues that grounding is concentrated in the middle layers and is primarily implemented by attention heads that aggregate environmental information into the prediction of corresponding linguistic tokens. The paper also extends the analysis to image-grounded dialogue and includes a case study on LLaVA, suggesting that related mechanisms may also appear in more realistic vision-language models.

**Compliance With Llm Reviewing Policy:**

Affirmed.

**Final Justification:**

The rebuttal clarified the authors' intended scope and provided some helpful additional robustness evidence, but it did not resolve my main concern, which is that the paper's central claim is framed more broadly than the evidence supports. The paper shows that environmental or visual context improves prediction of corresponding linguistic tokens, and that some components are causally important for that effect. That is an interesting result. However, this is not yet enough to justify the stronger claim suggested by the title and framing—that the paper demonstrates *where symbol grounding emerges*, or that it identifies *the underlying mechanisms*.

**Definition of symbol grounding.**

I am not persuaded by the authors' response on definition. As stated, their notion of grounding risks being too weak or permissive, since on that view many ordinary dependencies between a non-linguistic input stream and linguistic prediction could count as grounding. That does not sufficiently distinguish grounding from generic cross-modal predictive conditioning. In my view, the paper needed to more clearly define and justify the sense of "symbol grounding" at issue.

**Limited mechanistic evidence.**

This issue is connected to the limits of the mechanistic evidence. The tuned-lens, saliency, and intervention results are suggestive and point to plausible components, but they still provide only a partial account. Beyond the correlational evidence from the tuned-lens and saliency analyses, only the ablation experiments provide **causal** evidence that certain attention heads are important. However, it remains unclear what precise functional role these heads play, what representations they operate over, or how information flows through the relevant circuit. Establishing that kind of claim would likely require stronger causal analyses, such as activation patching and related methods used in recent mechanistic interpretability work (e.g., [1–4]). As a result, the current evidence supports the claim that certain heads are relevant to the behavior more than it supports a broader claim about the mechanism underlying the emergence of symbol grounding.

Relatedly, the co-occurrence analysis is useful in ruling out a simple frequency-based account, but divergence from co-occurrence alone does not establish that the model has learned grounded or structured representations in the stronger sense required for the paper's central claim.

Overall, I think the paper contains interesting analyses and meaningful empirical results, but in its current form, it establishes context-conditioned multimodal prediction more convincingly than it establishes symbol grounding, and it localizes relevant components more convincingly than it explains the mechanism of grounding itself. For that reason, I remain at weak reject.

[1] Todd et al. Function Vectors in Large Language Models. ICLR 2024.

[2] Feng & Steinhardt. How do Language Models Bind Entities in Context? ICLR 2024.

[3] Wu et al. How Do Transformers Learn Variable Binding in Symbolic Programs? ICML 2025.

[4] Assouel et al. Visual symbolic mechanisms: Emergent symbol processing in vision language models. ICLR 2026.

**Key Questions For Authors:**

1. Do the authors have any hypothesis for why unidirectional LSTMs fail in this setting? Is the limitation mainly due to long-context processing, representational capacity, or some more specific architectural constraint?

2. Could the authors clarify how the saliency thresholds were chosen, or how robust the gather / aggregate head results are to these threshold choices?

**Limitations:**

Given the current evidence, it would be more appropriate to frame the mechanistic account as a hypothesis supported by preliminary results rather than a fully established explanation. Similarly, the claims around "grounding" would benefit from more cautious framing or a clearer delineation of the precise sense in which the work demonstrates grounding.

**Strengths And Weaknesses:**

### Strengths

- The topic of symbol grounding in AI models is a timely and important one, which is of broad interest to multimodal learning and interpretability research.
- The paper goes beyond behavioral evidence by combining analyses of model behavior, training dynamics, and internal mechanisms, which makes the contribution more substantial. The comparison across architectures also provides a fuller picture.
- I also appreciated the attempt to extend the analysis to a more realistic setting.

### Weaknesses

- I have some reservations about the framing around "symbol grounding." The evidence convincingly shows that the models exploit environmental context to improve token prediction, but this seems broader and weaker than the notion of "symbol grounding" that the title and framing may suggest. In particular, the grounding metric based on surprisal is indirect and fairly generic, so the current results seem to establish contextually informed prediction more strongly than genuine symbol grounding.
- The mechanistic analyses are suggestive but still incomplete. The tuned-lens results mainly indicate when the model begins to favor the target token, rather than how that prediction is computed or whether the symbol is truly bound to the grounded environment. The attention analyses point to plausible components involved in the effect, but they read more as a partial account than a full explanation of the underlying computation.
- The causal intervention results on aggregate heads are promising, but they do not yet establish whether these heads are necessary or sufficient for the behavior. Although ablating them produces significant effects, it remains unclear whether the same function might be realized through multiple pathways, or whether these heads alone would be sufficient to support the behavior.

---

> ### Author Rebuttal · Authors · 2026-03-31
>
> We thank the reviewer for the thoughtful and detailed review. We hope the following clarifications and additional experiments address the reviewer's concerns.
>
> ## Weakness 1: Framing around "symbol grounding"
> We thank the reviewer for the thoughtful comments. We believe there may have been a misunderstanding regarding the purpose of our minimal testbed, which only serves as a noise-free proxy to help understand how grounding is implemented in VLMs, and the non-symbolic nature of "visual tokens" in VLMs. We kindly refer the reviewer to the response to Reviewer B8qm "Weakness 1 & Weakness 3: Relationship to symbol grounding" for details.
>
> ## Weakness 2: "The mechanistic analyses are incomplete"
> Our account of the emergence of grounding does not rely on any single method in isolation, but on the  **consistency across independent analyses** (§5.1–§5.4): (i) tuned-lens trajectories identify when predictions become environment-sensitive, (ii) attention-saliency traces show the emergence of relevant pathways during training, and (iii) head-level causal interventions demonstrate that specific components influence grounding behavior.
>
> While each method provides a "partial account," they independently reveal the same phenomenon: grounding is centrally encoded by aggregate heads in mid layers. This is not a single-method observation but a cross-validated mechanism.
>
> ## Weakness 3: Necessity and sufficiency of aggregate heads
>
> The reviewer suggests that the ablation experiment do not yet establish whether these heads are necessary or sufficient for the behavior. Our intervention experiments on aggregate heads consistently and significantly increases surprisal compared to controls confirms that aggregate heads are **at least causally necessary** for the observed grounding behavior.
>
> Regarding sufficiency, the reviewer doubts "the same function might be realized through multiple pathways" and "whether these heads alone would be sufficient to support the behavior". We observed that when the identified aggregate heads are ablated during evaluation, models re-route information and form new aggregate heads in previously non-aggregate heads. We did not include this result because: (i) it expands beyond the scope of this paper, and (ii) it warrants an analysis of its own. Importantly, this re-routing behavior underscores the crucial role of the aggregation mechanism itself. We will incorporate additional discussion and clarification in the future version of this paper.
>
>
> ## Question 1: Architectural Differences
> Current results suggest that symbol grounding emerges in Transformers and SSMs but not in LSTMs. We hypothesize that grounding depends on an explicit attention-like mechanism over past states, which allows a model to "look back" at the visual/environmental context and retrieve relevant information there. Transformers and SSMs provide this capability, whereas unidirectional LSTMs face a structural limitation: they must compress the entire history into a single fixed-size hidden state, which creates a bottleneck for effective grounding.
>
> ## Question 2: Sensitivity to Saliency Thresholds
> We thank the reviewer for bringing this up. The saliency thresholds were empirically selected to capture heads exhibiting the typical aggregation patterns observed in saliency maps. Regardless, our intervention results are robust to different choices:
>
> 1. In our VLM setting, we defined aggregate heads with two different thresholds, 0.7 and 0.9 (Figure 7, left). Causal interventions on heads identified at both thresholds consistently yielded significantly higher surprisals compared to controls. This shows that aggregation is a general mechanism, rather than an artifact sensitive to a specific operational threshold.
> 2. For our minimal testbed (CHILDES), we have run additional causal intervention experiments on a new threshold (20%):
>    + step 20000 -- 5.88 (aggregate Interv. Sps) vs 4.74 (aggregate Ctrl. Sps.) 4.59 (gather Interv. Sps.) 4.77 (original)
>    + step 10000 -- 6.041 (aggregate Interv. Sps) vs 5.279 (aggregate Ctrl. Sps.) 5.189 (gather Interv. Sps) 5.30 (original)
>    + step 5000 -- 6.584 (aggregate Interv. Sps) vs 6.395 (aggregate Ctrl. Sps.) 6.374 (gather Interv. Sps) 6.38 (original)
>
> The consistent surprisal gap (between aggregate head intervention group surprisal and its control group surprisal) across thresholds confirms that the operational definition does not fundamentally change the identification of the grounding mechanism.

---

> > ### Author Rebuttal · Reviewer_UYs7 · 2026-04-03
> >
> > Thank you for the detailed rebuttal. I appreciate the clarifications and the additional robustness results, especially regarding the saliency thresholds and the intended role of the minimal testbed as a controlled proxy for studying grounding mechanisms.
> >
> > That said, I still have some reservations that are only partially resolved.
> >
> > **Framing around symbol grounding.**
> >
> > As I understand it, the core empirical result is that matched environmental context improves prediction of the corresponding linguistic token, as measured through surprisal. That is an interesting and meaningful result, but it remains somewhat broader and weaker than the notion of "symbol grounding" suggested by the current framing. In other words, I now better understand the authors' intended interpretation, but I still think the claims would benefit from more careful wording about the precise sense of grounding that is demonstrated.
> >
> > **Mechanistic analyses.**
> >
> > I agree that the paper does not rely on any single analysis in isolation, and there is convergence across tuned-lens trajectories, saliency patterns, and head ablations. At the same time, I still view this as a partial mechanistic account rather than a complete one. The current evidence is suggestive about where the effect may emerge and which components may be involved, but it does not yet fully establish how the computation is implemented, what representations these components operate over, or whether the proposed account fully captures the relevant pathways. I think this matters in light of the grounding claim the paper aims to make. For example, it remains unclear whether the model learns a structured mapping between environmental and linguistic representations, rather than a more task-specific predictive dependency.
> >
> > **Necessity and sufficiency.**
> >
> > More specifically, I view the tuned-lens and saliency results as informative but primarily correlational: they help localize where environment-sensitive prediction begins to appear and which pathways are plausibly involved, but they do not by themselves establish that these components are causally responsible for the behavior. The intervention results are therefore especially important here, but the current ablation study, without stronger isolation or patching-based analyses, supports the claim that the identified aggregate heads play a causal role more than it establishes that they are necessary or sufficient for the observed behavior. The re-routing mechanism mentioned in the rebuttal is interesting and potentially important, but since it is not part of the current paper, I think the strongest supported claim remains that aggregate heads are relevant components, rather than that they fully characterize it.
> >
> > **Architectural differences.**
> >
> > Regarding the architectural differences, I appreciate the hypothesis about the hidden-state bottleneck in unidirectional LSTMs. It would be helpful to clarify whether the authors view this as a fundamental architectural limitation, or whether they believe similar behavior might emerge in sufficiently large LSTMs.
> >
> > Overall, the rebuttal improves my view of the paper by clarifying the intended scope and by providing additional robustness evidence. However, my main concerns remain: I still think the paper would benefit from more cautious framing around "symbol grounding," and I still see the mechanistic results as suggestive and valuable, but not yet a more established account of the phenomenon.

---

> > > ### Author Response · Authors · 2026-04-07
> > >
> > > We appreciate the reviewer's thoughtful response. We address the remaining concerns below:
> > >
> > > ### Framing around symbol grounding
> > > The reviewer mentioned
> > > > The core empirical result is that matched environmental context improves prediction of the corresponding linguistic token, as measured through surprisal ... but it remains somewhat broader and weaker than the notion of "symbol grounding" suggested by the current framing.
> > >
> > > Since this is our last possible response to the reviewer, we would like to note that according to Hanard (1990), the "symbol grounding" problem refers to "how can the meanings of the meaningless symbol tokens, manipulated solely on the basis of their (arbitrary) shapes, be grounded in *anything but other meaningless symbols*?" In this work, we use visual signals as the operational definition of ground (i.e., "*anything but other meaningless symbols*" by Harnad), and use the two-stream (`<ENV> <LAN>`) token mechanism to draw intuition from. Our causal intervention analysis has also suggested that the mechanism is beyond correlational.
> > >
> > > That said, if the reviewer could clarify their definition of "the precise sense of grounding", we would be happy to refine our terminology in the revision of this paper.
> > >
> > > ### Mechanistic analyses & Necessity and sufficiency
> > > The reviewer mentioned
> > > > It does not yet fully establish how the computation is implemented
> > >
> > > and
> > >
> > > > I think the strongest supported claim remains that aggregate heads are relevant components, rather than that they fully characterize it.
> > >
> > > We would like to note that we have never claimed that this paper is a full characterization of symbol grounding---establishing a "full" account that explains every exact computation in a large-scale neural network would be a long-standing and perhaps infeasible challenge for the field. Our goal is to identify functionally central components that implement grounding, rather than to "fully" explain every component and representation, or to capture all the relevant pathways.
> > >
> > > Evidence in Table 2 (last row) supports our goal: ablating these aggregate heads (only 4.7% of all the attention heads) leads to a significant surprisal increase, whereas ablating an equivalent number of control heads has no effect. This demonstrates that these heads are not merely "relevant" but are causally accoutable to the behavior.
> > >
> > > > ...it remains unclear whether the model learns a structured mapping between environmental and linguistic representations, rather than a more task-specific predictive dependency.
> > >
> > > If we understand the reviewer correctly about what "task-specific predictive dependency" refers to, we believe our co-occurrence analysis addresses this. We find that:
> > >
> > > - Early in training, information gain aligns with surface-level co-occurrence (a predictable baseline).
> > > - Later in training, information gain diverges sharply from any co-occurrence-based predictor, especially for abstract or relational tokens where naive frequency cues fail.
> > >
> > > During training, the model moves from exploiting superficial statistical co-occurence to forming context-sensitive, structured representations, which suggests that this shortcut (i.e., what we understand as "task-specific predictive dependency" mentioned by the reviewer) cannot fully explain our results.
> > >
> > >
> > > ### Architectural differences
> > >
> > > We view the hidden-state bottleneck in unidirectional LSTMs as a fundamental architectural limitation. Since the aggregate mechanism is the most important part that enables symbol grounding, we believe grounding is unlikely to emerge in LSTMs without position-based attention. We will leave this investigation as part of the future work.

---

### Official Review · Reviewer_B8qm · 2026-03-12

**Soundness:** 2
**Presentation:** 3
**Significance:** 3
**Originality:** 2
**Overall Recommendation:** 4
**Confidence:** 3

**Summary:**

The paper explores how different model architectures address the symbol grounding problem, originally formulated by Harnad (1990). The authors train several models from scratch on a range of corpora, including CHILDES and a visual dialogue corpus. They manipulate the contextual information available to the mode, providing either context that is relevant or irrelevant to the utterance being predicted. The experiments show Transformers and SSMs exhibit a sharp drop in surprisal in the relevant-context condition, while LSTMs do not show this behavior. The authors also introduce mechanistic analyses to identify the attention heads that play a key role in enabling this capability.

**Compliance With Llm Reviewing Policy:**

Affirmed.

**Final Justification:**

My concerns have been solved.

**Key Questions For Authors:**

See weaknesses.

**Limitations:**

No.

**Strengths And Weaknesses:**

### Strength
1. The authors train the models from scratch in a careful and controlled manner, which enables more conclusive interpretations of the results.
2. The comparison across different architectures is particularly informative, providing a clear illustration of the importance of architectural binding mechanisms.

### Weakness
1. This paper does not seem to have much of anything to do with symbol grounding.
2. Based on the new word tokens, do you retrain the tokenizers? How big is the new tokenizer? Is it for all tokens to have both <env> and <lan>?
3. The setup only relies on two linguistic tokens that denote the same object. The “env” tokes are still merely symbols and lack any grounding. Therefore, the model’s task seems to learn a mapping between (A<ENV>) and (A<LAN>), rather than grounding symbols in real-world referents.
4. For the image-text setting, why do you remove the mlp projector from the model?
5. Like the example in Figure 1b, how to evaluate the performance just based on one patch from the whole image?
6. Missing related work:
    - Do Vision and Language Models Share Concepts? A Vector Space Alignment Study. TACL, 2024.
    - The Platonic Representation Hypothesis, ICML 2024

---

> ### Author Rebuttal · Authors · 2026-03-31
>
> ## Weaknesses 1 & 3: Relationship to symbol grounding
>
> The reviewer says "This paper does not seem to have much of anything to do with symbol grounding". We believe there may have been a misunderstanding regarding the purpose of our minimal testbed, the technical implementation and non-symbolic nature of "visual tokens" in vision-language models, or both.
>
> > The setup only relies on two linguistic tokens that denote the same object.
>
> First, we would like to clarify that in our CHILDES setup (minimal testbed), there are not "two linguistic tokens", but "one **enviornment** token" and "one **linguistic** token".
> The purpose of our minimal testbed is to provide a (nearly) fully controlled, noise-free proxy that helps understand how grounding is implemented in realistic vision-language models (VLMs). Hypotheses derived under the testbed are subsequently validated in realistic VLMs, and all major findings in the testbed hold in the VLM settings.
>
> Crucially, in the VLM setting, the model is indeed "grounding symbols in real-world referents", as the visual inputs (so-called "visual tokens") are nonsymbolic. The "visual tokens" are continuously valued vectors produced by a pretrained neural image encoder, each derived from a square-shaped patch of the input image. Therefore, **a more accurate perspective is that $n$ visual tokens of dimension $d$ constitute an $n \times d$ tensor that represents visual features of an image, rather than $n$ discrete symbolic tokens** (as mentioned in footnote 2, page 4). In our work, we use DINOv2 (a vision-only self-supervised image encoder; Oquab et al., 2024) as the visual feature extractor, which outputs high-quality and faithful representation of the raw perceptral input. This design choice avoids a major concern associated with CLIP ViT (the most common practice in VLMs; Radford et al., 2021) that its image representations may inherit symbolic or linguistic regularities introduced during vision-language pretraining.
>
> Given the clarification above, we believe our work clearly conntects to explaining how VLMs implement the symbol grounding mechanism. Should the reviewer have any remaining concerns along this line, we would appreciate it if they could elaborate on why they hold the position that "this paper does not seem to have much of anything to do with symbol grounding".
>
>
> ### References
> [1] Oquab et al. 2024. DINOv2: Learning Robust Visual Features without Supervision. In TMLR.
>
> [2] Radford et al. 2021. Learning transferable visual models from natural language supervision. In ICML.
>
> ## Weakness 2: Details about tokenizers
> We clarify that our tokenizer was trained from scratch (not "retrained"), using a straightforward word-level approach that splits sentences based on spaces. For example, box:<ENV> and box:<LAN> are treated as distinct tokens with separate integer indices; thus, any connection between them must be learned during training.  The vocabulary size is 89,057 (built on ~22M token training corpus). Only 9454 words have both `<ENV>` and `<LAN>` variants, as they appeared in both the environmental descriptions and the dialogue. (We assume the [object Object] artifacts in the review refer to our `<ENV>` and `<LAN>` tags.)
> ## Weakness 4: Reasons for removing the mlp projector
> Unlike the standard LLaVA (Liu et al., 2023) aligning a frozen pre-trained CLIP encoder with a pre-trained LM, we trained the LM backbone (GPT-2) from scratch. We use frozen DINOv2 as our vision encoder and feed its representations directly into the GPT-2 model. Since the DINOv2 output dimension matches the GPT-2 hidden dimension (768), an MLP projector is architecturally unnecessary.
> Additionally, we hypothesize that training from scratch allows the GPT-2 to naturally acquire vision-language alignment; experiments show that omitting the MLP projector yields slightly better performance.
>
> ## Weakness 5: Evaluation protocal in image-grounded setting
> We respectfully clarify that we do not "evaluate the performance just based on one patch".
>
> As described in Sections 3.2 and 4.3, we use *grounding information gain* to measure how much information the matched ground provides for predicting the linguistic form, compared to a mismatched one. In the VLM setting, the matched ground consists of all ViT-derived visual tokens within the bounding box (or ground-truth mask region), coverin the entire referent (Table 1). Mismatched contexts are generated with Stable Diffusion 2, and we always report the surprisal gap between match and mismatch conditions.
>
> As for Figure 1(b), it is an illustration of an aggregate head in a real-world VLM. The highlighted patch represents a particularly strong information flow from this patch (within the bounding box) to the current position, exemplifying the aggregation mechanism.
>
> ## Weakness 6: Gaps in related work
> We thank the reviewer for highlighting these relevant works. We will incorporate them in the camera-ready version.

---

> > ### Author Rebuttal · Reviewer_B8qm · 2026-04-03
> >
> > Thanks for the authors' clarification. My concerns have been solved.

---

### Official Review · Reviewer_bdw5 · 2026-03-13

**Soundness:** 3
**Presentation:** 3
**Significance:** 3
**Originality:** 3
**Overall Recommendation:** 5
**Confidence:** 5

**Summary:**

This paper studies how symbol grounding emerges in VLMs. The authors propose a controlled evaluation framework that tracks grounding throughout training using a synthetic testbed derived from the CHILDES child-directed speech dataset and an image-grounded dialogue setup. Experiments across Transformers, state-space models, and LSTMs show that grounding emerges during training in Transformers and Mamba but not in unidirectional LSTMs.

**Compliance With Llm Reviewing Policy:**

Affirmed.

**Final Justification:**

The authors have addressed most of my concerns and i will keep my original score.

**Key Questions For Authors:**

1. The experiments use a vocabulary of only 100 nouns. What was the motivation behind restricting the vocabulary to this size?
2. The paper shows that Transformers and Mamba exhibit grounding but LSTMs do not. Could you elaborate on which architectural properties are most critical for enabling grounding?

**Limitations:**

yes

**Strengths And Weaknesses:**

Strengths:
1. They have a strong and controlled experimental setup. For example, The authors train models from scratch, which gives them full control over the training dynamics and removes effects from pretrained knowledge. They also evaluate grounding from multiple perspectives, including behavioral analysis, training dynamics, mechanistic interpretability, and causal interventions.
2. The paper is easy to follow, with well-designed figures, tables, and visualizations that help explain both the experimental setup and the results.
3. I also liked their cross-architecture comparison with transformers, LSTMs and state space models.

Weakness:
1. Although the authors briefly test on LLaVA, the main mechanistic and training-dynamics experiments are conducted on relatively small models, leaving open questions about whether the same mechanisms scale to large modern VLMs.

---

> ### Author Rebuttal · Authors · 2026-03-31
>
> We thank the reviewers for their thoughtful comments and the positive assessment of our work.
> ## Weakness: Generalization to Full-Scale VLMs
> We thank the reviewer for pointing out this important direction, which we share completely as discussed in Section 6. Challenges for extending our analysis to full-scale VLMs like LLaVA is that they used CLIP-derived embeddings that already encode language priors for better performance, and that global information may be stored in tokens out of the object regions (known as registers). We completely agree that additional causal analysis on full-scale VLMs is an important direction of future work.
>
>
> ## Questions 1: Vocabulary Size
> Thank you for asking this question and we are happy to clarify:
>
> The training vocabulary is substantially larger than 100 words. It includes all word-level tokens present in the CHILDES training corpus, resulting in a vocabulary of approximately 89,000 tokens. The number “100” refers only to the subset of concrete nouns used in the evaluation stage.
>
> These 100 nouns were not selected randomly. We selected candidate nouns from the MacArthur–Bates Communicative Development Inventories (CDI) that also appear in the model’s vocabulary, and ranked them by the frequency of their environmental token (`<ENV>`) occurrences in the training data. We observed that the 100th-ranked noun has approximately 100 environmental token occurrences, and used this point as a cutoff to ensure that each evaluated noun had sufficient environmental exposure for meaningful analysis.
>
> ## Questions 2
> Current results suggest that symbol grounding emerges in Transformers and SSMs but not in LSTMs. We hypothesize that grounding depends on an explicit attention-like mechanism over past states, which allows a model to "look back" at the visual/environmental context and retrieve relevant information there. Transformers and SSMs provide this capability, whereas unidirectional LSTMs face a structural limitation: they must compress the entire history into a single fixed-size hidden state, which creates a bottleneck for effective grounding.

---

> > ### Author Rebuttal · Reviewer_bdw5 · 2026-04-03
> >
> > I am satisfied with their response and would maintain my positive score.

---

### Decision · Program_Chairs · 2026-04-30

**Decision:**

Accept (regular)

**Comment:**

Overall, this is a well-written paper exploring an interesting and potentially valuable topic area. Although overall the paper covers interesting ground, I would strongly suggest that the authors take seriously the concern of multiple reviewers that using a linguistic token to predict another token (even a visual one) is symbol grounding only in a relatively weak sense, as there is not a strong connection to a physically perceived environment. (This meta-review takes author(s)' concerns ont his front into account.) I recommend thinking about how to scope the claims made about the explanatory power of the work to ensure that this more typical sense of grounding is not confusing: "understand[ing] the emergence of symbol grounding in multimodal language models" is a significantly strong claim, and there are plenty of papers about physical grounding of language that it does not apply to. More generally, the work would be well served by thinking seriously about how to incorporate reviewers' suggestions.